# Artificial Intelligence in Nutrition and Dietetics: A Comprehensive Review of Current Research

**DOI:** 10.3390/healthcare13202579

**Published:** 2025-10-14

**Authors:** Gabriela Georgieva Panayotova

**Affiliations:** Division of Physiology, Department of Physiology and Pathophysiology, Medical University of Varna, 9002 Varna, Bulgaria; gabrielapanayotova95@gmail.com

**Keywords:** AI methods, nutrition technology, dietary assessment, personalized nutrition, machine learning

## Abstract

**Background/Objectives**: Artificial intelligence (AI) has emerged as a transformative force in healthcare, with nutrition and dietetics becoming key areas of application. AI technologies are being employed to enhance dietary assessment, personalize nutrition plans, manage chronic diseases, deliver virtual coaching, and support public health nutrition. This review aims to critically synthesize the current literature on AI applications in nutrition, identify research gaps, and outline directions for future development. **Methods**: A systematic literature search was conducted across PubMed, Scopus, Web of Science, and Google Scholar for peer-reviewed publications from January 2020 to July 2025. The search included studies involving AI applications in nutrition, dietetics, or public health nutrition. Articles were screened based on predefined inclusion and exclusion criteria. Thematic analysis grouped findings into six categories: dietary assessment, personalized nutrition and chronic disease management, generative AI and conversational agents, global/public health nutrition, sensory science and food innovation, and ethical and professional considerations. **Results**: AI-driven systems show strong potential for improving dietary tracking accuracy, generating personalized diet recommendations, and supporting disease-specific nutrition management. Chatbots and large language models (LLMs) are increasingly used for education and support. Despite this progress, challenges remain regarding model transparency, ethical use of health data, limited generalizability across diverse populations, and underrepresentation of low-resource settings. **Conclusions**: AI offers promising solutions to modern nutritional challenges. However, responsible development, ethical oversight, and inclusive validation across populations are essential to ensure equitable and safe integration into clinical and public health practice.

## 1. Introduction

Nutrition and dietetics play a central role in the prevention and management of non-communicable diseases such as obesity, type 2 diabetes, cardiovascular disease, and cancer [1]. As the burden of these conditions continues to rise globally, there is an urgent need for scalable, personalized, and evidence-based nutritional interventions [2,3]. At the same time, technological innovations—particularly those driven by artificial intelligence (AI)—are transforming the landscape of healthcare, offering novel tools for assessment, diagnosis, education, and behavioral change [4].

Beyond non-communicable conditions, nutritional status also shapes susceptibility to and recovery from communicable diseases [5]. Adequate energy and protein intake, together with sufficient micronutrients (e.g., vitamins A, D, C and B-complex; zinc, selenium, and iron), supports epithelial barrier integrity, innate and adaptive immunity, antibody production, and wound healing—thereby lowering infection risk and severity [6,7,8,9]. Conversely, protein–energy malnutrition and key micronutrient deficiencies increase morbidity and prolong convalescence, particularly in infants, older adults, and other vulnerable groups [10]. During active infection, tailored nutrition care (appropriate energy/protein targets, hydration, and micronutrient repletion as clinically indicated) can mitigate catabolism and functional decline, while public-health nutrition measures—breastfeeding support, complementary feeding guidance, food fortification and targeted supplementation, school meal programs, and alignment with water, sanitation and hygiene and immunization initiatives—help reduce incidence and improve outcomes at the population level [11,12].

AI, encompassing machine learning (ML), deep learning (DL), and natural language processing (NLP), has gained increasing attention in recent years for its capacity to analyze vast and complex datasets, recognize patterns, and generate tailored recommendations [13]. In the context of nutrition science, AI is being explored and applied across various domains, including dietary assessment, food recognition, personalized diet planning, chronic disease management, nutrition education, and even sensory analysis of food products [14,15,16]. These applications promise not only increased efficiency and accuracy but also the potential to democratize access to quality dietary advice—especially in underserved or resource-limited settings [17].

The emergence of generative AI and LLMs, such as ChatGPT 4.0, has further expanded the scope of AI in dietetics by enabling real-time interaction with users, simulating dietitian consultations, and supporting health literacy through personalized education [18,19]. Meanwhile, mobile apps, wearable sensors, and AI-enabled image recognition are redefining how individuals track their intake and modify their behavior [20]. Despite this progress, critical questions remain about the validity, safety, ethical implications, and long-term effectiveness of these technologies in real-world settings [21].

The aim of this review is to explore how AI is reshaping the field of nutrition and dietetics by enhancing both scientific understanding and practical delivery of dietary care. It seeks to capture the broader significance and transformative potential of AI technologies in empowering healthcare professionals, engaging patients, and improving population-level health outcomes. The review further aims to stimulate critical reflection on how nutrition science must evolve in response to these technological advancements.

## 2. Materials and Methods

### 2.1. Objective and Scope

The primary objective of this review is to systematically examine and critically synthesize peer-reviewed academic literature on the current applications of AI in nutrition and dietetics. The scope encompasses a wide range of use cases, including:AI-driven dietary assessment and food recognition;Personalized nutrition planning and metabolic prediction;Clinical decision support in chronic disease contexts;Generative AI and conversational agents in patient education;Mobile health applications and remote coaching;Ethical, regulatory, and professional practice considerations.

The six categories were chosen through an iterative thematic analysis of the included studies. They represent the domains where applications of AI in nutrition and dietetics were most frequently studied and where sufficient peer-reviewed evidence was available for meaningful synthesis. While other areas such as remote patient monitoring, women’s health, or rural health applications are important, they were either underrepresented in the literature retrieved during our review period or addressed only tangentially without a primary focus on nutrition. For this reason, the evidence was grouped into six categories that best capture both the breadth of AI applications and the depth of available evidence.

The review focuses on studies published between 2020 and 2025, integrating evidence from both clinical and community settings. Its structured approach aims to map existing knowledge, highlight methodological strengths and limitations, and identify future research priorities to guide responsible integration of AI technologies in the field.

### 2.2. Search Strategy

A comprehensive literature search was conducted across several scientific databases, including PubMed, Scopus, Web of Science, and Google Scholar. The search spanned publications from January 2020 to July 2025, ensuring the inclusion of recent developments in AI and its integration with nutrition science. Representative search terms included “Artificial Intelligence”, “Machine Learning”, “Deep Learning”, “Natural Language Processing”, “Large Language Models”, combined with “Nutrition”, “Dietetics”, “Dietary Assessment”, “Personalized Nutrition”, “Diet Planning”, “Malnutrition”, “Virtual Health Coaching”, and “Nutrition Education”.

### 2.3. Inclusion and Exclusion Criteria

Inclusion criteria were defined to ensure relevance and quality of the sources:Peer-reviewed journal articles, conference papers, and systematic reviews;Studies focused on the development, validation, or application of AI in nutrition science, dietetics, or public health nutrition;Articles written in English;Both clinical and non-clinical settings.

Exclusion criteria included:Editorials, opinion pieces, or commentaries without empirical data;Articles that addressed general AI in healthcare without specific mention of nutrition or dietetics;Studies with poor methodological quality (e.g., no evaluation/validation methods reported).

### 2.4. Data Extraction and Synthesis

To document study selection, a PRISMA-style flow was followed (Figure 1). The database search identified 136 records (PubMed [*n* = 57], Scopus [*n* = 35], Web of Science [*n* = 17], Google Scholar [*n* = 27]). After duplicate removal (*n* = 4), 132 records underwent title/abstract screening; 24 were excluded for irrelevance. 108 full texts were assessed, excluding 66 with reasons (e.g., not nutrition-specific [*n* = 18], no AI component [*n* = 6], non-empirical/editorial [*n* = 23], poor reporting of methods [*n* = 19]). 42 studies met all criteria and were included in the synthesis.

For each selected study, data were extracted regarding:AI technique used (e.g., supervised learning, deep learning, LLMs);Nutrition-related application (e.g., assessment, education, clinical support);Target population or health condition;Study design and setting;Key findings and performance metrics (e.g., accuracy, precision, user satisfaction);Limitations and future research suggestions.

The findings were grouped into six thematic categories, which form the structure of the following section:AI in dietary assessment;AI in personalized and clinical nutrition;Generative AI and conversational agents in nutrition advice;Mobile apps and virtual coaching;AI in global nutrition and public health;Ethical and professional implications of AI in dietetics.

## 3. AI Applications in Nutrition and Dietetics

### 3.1. AI for Dietary Assessment and Nutrient Tracking

One of the most prominent applications of AI in nutrition science lies in dietary assessment and nutrient tracking, where AI is used to estimate caloric and nutritional intake based on food images, natural language input, or user interaction with digital platforms [22].

A leading example in this domain is goFOOD^TM^ 2.0, an AI-powered dietary assessment tool that utilizes computer vision to identify foods and estimate portion sizes from photographs. The system combines deep learning models for food recognition and volume estimation, offering users immediate feedback on energy intake without the need for manual logging. Its real-time assessments have shown promise in preliminary validation studies [23].

In a comparative analysis, image-based AI dietary tools were assessed against registered dietitians and ground-truth data. Results demonstrated that although AI systems like goFOOD^TM^ and other deep learning-based apps can closely approximate expert estimations, discrepancies still exist, particularly in complex meals with mixed dishes, occlusions, or when portion size estimation is ambiguous [22,23].

A systematic review revealed that the accuracy of AI-based dietary assessment tools varies across platforms and food types, with common sources of error including lighting conditions, food presentation, and lack of standardized food databases [22]. Some studies emphasize the need for further training of models on culturally diverse food datasets and integrating user feedback loops to improve system performance over time [15,24].

Furthermore, recent research into smartphone-based applications using AI has confirmed their usability and potential in large-scale dietary monitoring. These tools not only reduce the burden of self-reporting but also enhance user engagement, especially when integrated with behavior change features like gamification or coaching [25,26].

However, challenges remain. These include:Variability in image quality affecting recognition accuracy;Insufficient database coverage for regional or homemade dishes;Difficulty estimating mixed meals or hidden ingredients;User compliance, especially in consistently photographing meals.

Despite these limitations, AI image recognition and nutrient estimation systems represent a significant leap toward automated, scalable, and real-time dietary assessment, particularly valuable for research, clinical nutrition, and personal health optimization [22]. Importantly, not all studies in this domain are of equal methodological rigor. For instance, validation trials of the goFOOD^TM^ system demonstrated moderate agreement with registered dietitians under real-world conditions [23], whereas several exploratory tools lacked external validation and used limited or homogeneous datasets [22]. This highlights the need to distinguish between well-validated systems with clinical potential and preliminary applications that remain proof-of-concept.

### 3.2. AI-Driven Personalized Nutrition and Disease Management

AI technologies are currently employed in managing conditions such as Type 2 Diabetes Mellitus (T2DM), obesity, and cardiovascular diseases through data-driven personalization of dietary strategies [14]. These models leverage individual physiological and behavioral data to inform dietary recommendations, often improving adherence and clinical outcomes [24]. Additionally, virtual dietetic assistants and machine learning–based planning tools are increasingly integrated into clinical workflows, providing decision support and streamlining patient counseling [13].

#### 3.2.1. AI Models in T2DM, Obesity, and Cardiovascular Health

One prominent application is the use of AI-driven systems to support dietary management in T2DM. A study by Sun et al. introduced an AI dietitian that integrates large language models and image recognition tools to recommend personalized meal plans and analyze nutritional intake for T2DM patients. This system demonstrated high performance in dietary analysis and potential clinical utility in preclinical validation [27]. In contrast, randomized controlled trials of AI-driven health coaching interventions reported significant improvements in diet and physical activity adherence, thus providing higher-quality evidence than feasibility or simulation-based studies [28,29].

In the domain of weight management, several studies have evaluated the effectiveness of AI-generated diet plans. While these tools can generate diets that meet macronutrient targets, their qualitative evaluations revealed variability in accuracy, safety, and personalization, indicating the need for professional oversight when using such tools in real-world interventions [27,30,31].

Additionally, AI has been explored for its potential in cardiovascular health management. By leveraging patient data such as lipid profiles, body mass index, and lifestyle behaviors, machine learning models can generate risk scores and recommend dietary interventions that align with evidence-based cardiovascular prevention strategies [29].

#### 3.2.2. Dietitian-Assistive Chatbots and Virtual Coaches

The rise in AI-powered chatbots and virtual coaches presents new avenues for scalable, round-the-clock dietary counseling. In a proof-of-concept study, an AI virtual health coach delivered a 12-week diet and physical activity intervention that yielded significant improvements in participants’ behavior, including fruit and vegetable intake and physical activity levels. Participants rated the experience as engaging and informative, supporting the potential of AI systems to extend the reach of human dietitians [28].

Another investigation compared different AI chatbots, including ChatGPT, Bing Chat, and Google Bard, in providing dietary advice. While all systems could deliver basic nutritional guidance, inconsistencies in accuracy, reproducibility, and evidence sourcing highlighted current limitations in their clinical reliability [32].

#### 3.2.3. Machine Learning for Dietary Plan Creation and Metabolic Prediction

Machine learning models are increasingly being trained on large datasets comprising biometric, genomic, and lifestyle variables to predict individual metabolic responses to foods [27]. These models can enable highly personalized dietary plans that optimize glycemic control, weight loss, and lipid profiles. While these applications are still emerging, their promise lies in transitioning from generalized to truly individualized nutrition therapy [2,27].

Furthermore, digital nutrition apps that incorporate AI decision-support algorithms are gaining popularity for real-time meal tracking, health goal setting, and user engagement [33]. Mixed-methods evaluations show that users find these tools beneficial for fostering healthier eating habits, although long-term adherence and clinical outcomes remain active areas of research [34].

### 3.3. Generative AI and Conversational Agents in Nutrition

The emergence of generative AI, particularly LLMs like ChatGPT, has opened new avenues for providing real-time, personalized dietary advice, nutrition education, and patient engagement [24]. These systems are increasingly being tested for their feasibility and accuracy in nutrition counseling, showcasing potential to support both professionals and the general public [15].

#### 3.3.1. ChatGPT and LLMs for Dietary Advice

ChatGPT and other LLMs have been evaluated for their ability to deliver personalized diet-related recommendations. In a study assessing ChatGPT’s capabilities in providing dietary advice, researchers found that while it offered generally relevant suggestions aligned with dietary guidelines, it lacked contextual nuance in individual medical conditions and did not always provide accurate citations or nutrient data [35]. Another analysis showed ChatGPT could offer appropriate advice for general healthy eating, but struggled with more specific or clinical queries, highlighting the need for human oversight in clinical applications [36].

#### 3.3.2. Comparison of Chatbot Accuracy, Consistency, and Safety

Multiple AI chatbots, including ChatGPT, Google Bard, Bing Chat, and others, have been benchmarked for their accuracy, completeness, reproducibility, and consistency in responding to nutrition-related queries. Results from a comparative evaluation revealed notable variability across platforms [32]. ChatGPT, for example, demonstrated higher consistency and more accurate calorie estimations than some competitors but was still prone to occasional factual errors and omitted citations [35]. These findings underscore the current limitations of unsupervised AI agents in delivering medically safe and reliable nutrition advice without domain-specific training or regulatory vetting [24].

#### 3.3.3. Use in Education and Patient Communication

Generative AI has also been explored as a pedagogical tool in nutrition education. A recent study by Chen and Liou proposed integrating ChatGPT into communication training for dietetics students, demonstrating how AI-generated simulations could enhance students’ confidence and reflective learning in counseling scenarios [37]. The virtual dialogues offered by ChatGPT helped students practice empathic communication and improve feedback interpretation, with high engagement and positive reception from participants [38].

Furthermore, conversational agents can support patient-centered communication by simplifying complex nutritional guidelines and maintaining engagement in long-term dietary interventions. Their 24/7 availability and capacity for personalization position them as valuable adjuncts in digital health interventions [39].

### 3.4. AI in Public and Global Health Nutrition

AI has begun to play a pivotal role in addressing global nutrition challenges, particularly in low-resource environments where traditional healthcare infrastructure is lacking. AI tools are increasingly being leveraged to enhance malnutrition screening, optimize resource allocation, and inform policy decisions related to public health nutrition [14].

#### 3.4.1. AI Tools for Malnutrition Screening in Resource-Limited Settings

In regions burdened by undernutrition and limited access to trained healthcare professionals, AI offers a scalable solution to improve early detection and intervention [40]. Machine learning algorithms have been employed to assess nutritional status using easily accessible parameters such as anthropometric data, dietary intake surveys, and demographic information [16]. A systematic review highlighted that AI models have shown promising results in predicting malnutrition risks, enabling timely intervention and reducing the burden on health systems in resource-constrained environments [41].

These models can process complex datasets to identify patterns and risk factors that might be missed through conventional screening methods. Additionally, AI-based mobile applications and diagnostic platforms can operate offline or with minimal connectivity, further enhancing their applicability in remote and underserved areas [42].

#### 3.4.2. Global Health Implications and Policy Integration

Beyond diagnostics, AI is influencing nutrition-related policy-making through its capacity to analyze large-scale public health datasets and simulate intervention scenarios. According to a review on AI and global health, AI technologies can help bridge inequalities in access to care and improve food and nutrition security by supporting evidence-based decisions at national and international levels [43,44].

The World Health Organization (WHO) and other international bodies are exploring how digital health innovations, including AI, can be ethically integrated into global health strategies. However, challenges remain related to algorithmic bias, data sovereignty, infrastructure gaps, and workforce readiness [21,45]. Effective policy integration requires multi-sector collaboration and adherence to global ethical standards to ensure that AI interventions are equitable, safe, and context-appropriate [21].

### 3.5. AI for Sensory Science and Food Innovation

AI is emerging as a powerful tool in sensory science and the field of food innovation, especially in decoding and designing flavor profiles based on consumer preferences and nutritional needs [46].

AI models, particularly machine learning algorithms, are being applied to analyze complex chemical compositions of foods and correlate them with sensory descriptors such as taste, aroma, and texture [47]. These systems leverage large databases of flavor molecules and sensory data to predict how combinations of ingredients will be perceived by consumers. This has direct applications in the development of healthier alternatives that retain desirable sensory characteristics [48].

For example, advanced models can simulate flavor profiles and recommend ingredient substitutions that lower sugar or sodium content without compromising palatability, addressing public health goals through personalized taste-based nutrition [49].

A key innovation area is using AI to link flavor perception with nutrient intake patterns and preferences. By analyzing individual responses to different flavors and food textures, AI can support the creation of more appealing and nutritious products tailored to specific populations, such as older adults, individuals with reduced taste sensitivity, or patients with dietary restrictions [48,49].

These advancements mark a significant shift from traditional sensory analysis to a more data-driven, predictive, and personalized approach to food development, with implications for industry, public health, and consumer experience (Table 1).

## 4. Ethical, Practical, and Professional Considerations

### 4.1. Ethical Challenges

The integration of AI into nutrition and dietetics introduces important ethical concerns that must be addressed to ensure safe, equitable, and trustworthy application (Table 2).

One of the most pressing issues is bias in training data. AI models trained on datasets that lack diversity—whether in terms of demographic, geographic, or socioeconomic factors—risk perpetuating and even amplifying health disparities [14,50]. For instance, nutrition-focused AI applications trained primarily on data from high-income countries may fail to provide accurate guidance in underrepresented populations or low-resource settings [14,51].

Transparency and explainability represent another ethical cornerstone. Many AI systems operate as “black boxes,” producing outputs without interpretable reasoning. This lack of algorithmic transparency impairs users’ and professionals’ ability to understand, trust, and validate AI-generated dietary recommendations [52]. Calls for explainable AI (XAI) have thus become increasingly important in the healthcare and nutrition sectors [53].

In addition, data privacy and security are crucial considerations, particularly for applications handling sensitive health and dietary data [54]. Ensuring compliance with data protection laws such as the GDPR and implementing robust anonymization and encryption protocols are essential to preserve user trust and protect against breaches and misuse [55].

### 4.2. Professional Roles and AI Integration

A central question raised by the advancement of AI in nutrition is: Will AI replace dietitians? While some early chatbot-based experiments and generative language models like ChatGPT have demonstrated the ability to provide basic nutrition information and even personalized diet plans, studies consistently reveal limitations in the accuracy, safety, and contextual relevance of these systems when compared to human professionals [19,35,56].

Rather than full replacement, the emerging consensus suggests a task-shifting and collaboration model. AI can efficiently automate routine assessments, meal tracking, and education, thereby freeing dietitians to focus on high-level clinical decision-making and complex patient counseling [16,21]. This synergy positions AI as a supportive tool within a multidisciplinary framework rather than as a substitute for professional expertise.

This evolution, however, necessitates education and training for current and future dietitians to work effectively with AI technologies. Nutrition professionals will need to acquire digital literacy and basic AI fluency to interpret AI outputs, detect potential errors, and communicate limitations to patients [14,57]. Integrating such competencies into nutrition education programs and continuing professional development will be essential to enable meaningful collaboration with AI systems [58].

## 5. Evaluation of AI Tools in Practice

The practical utility of AI in nutrition and dietetics depends heavily on real-world evaluations of usability, validity, and clinical effectiveness [16]. A growing body of literature, including mixed-methods research, randomized controlled trials (RCTs), and pilot studies, has begun to assess the performance of AI-driven nutrition applications in diverse populations.

### 5.1. Usability and Acceptance

Patient and professional acceptance is a critical determinant of successful AI integration in dietary practice. Studies show that users generally perceive AI nutrition apps as convenient, time-saving, and engaging, especially when visual or gamified components are included [59,60,61]. For instance, a mixed-methods study by Chew et al. (2024) found that participants using an AI-assisted food tracking app reported high levels of satisfaction, noting the tool’s ease of use, personalized feedback, and motivational features [62]. However, clinicians voiced concerns regarding data accuracy and ethical transparency [50].

### 5.2. Validity, Accuracy, and Reproducibility

AI-generated dietary assessments and recommendations must meet rigorous standards of validity and consistency to be trusted in clinical and public health settings. A pilot study evaluating the goFOOD^TM^ system found that its image-based dietary estimates showed moderate agreement with dietitian assessments, although performance varied based on food type and lighting conditions [23]. Similarly, a comparison of multiple AI chatbots revealed discrepancies in nutrient advice, with some tools offering unsafe or scientifically unsupported recommendations, emphasizing the need for robust validation protocols [32].

Reproducibility remains a challenge. AI models may generate varying results across platforms or versions due to differences in algorithms, training datasets, or natural language prompts. Standardization of input-output formats and clinical testing frameworks is therefore crucial for dependable deployment [63,64].

To address these challenges, several solutions could be pursued. First, the adoption of standardized input formats (e.g., harmonized food image capture protocols, structured dietary logs) would reduce variability across platforms and improve reproducibility. Second, the creation of reference nutrient databases with international coverage—including culturally diverse foods and preparation methods—would strengthen generalizability. Third, developers should adopt transparent reporting frameworks such as Consolidated Standards of Reporting Trials—AI (CONSORT-AI) or MINimum Information for Medical AI Reporting (MINIMAR), ensuring that algorithms, datasets, and performance metrics are consistently disclosed. Fourth, federated learning approaches could enable models to be trained on data from multiple institutions or countries without compromising privacy, thereby enhancing both accuracy and equity. Finally, establishing independent validation consortia—similar to those in radiology or genomics—would allow for cross-platform benchmarking and external validation of AI nutrition tools in real-world settings. Together, these measures could provide a pathway toward standardized, reliable, and clinically acceptable AI applications in nutrition and dietetics.

### 5.3. Mixed-Methods and RCT Evidence

Robust evidence on AI effectiveness is still emerging. In one of the few RCTs available, an AI-enhanced nutrition app significantly improved users’ consumption of fruits and vegetables and reduced intake of sugary beverages over a 3-month period compared to a control group [65]. Mixed-methods approaches, combining quantitative metrics with qualitative interviews, offer nuanced insight into both behavioral outcomes and user experiences, strengthening the evidence base for AI-enabled nutrition interventions [2]. Despite promising findings, most studies remain small-scale or short-term, underscoring the need for larger, multicenter trials that assess both clinical outcomes and long-term adherence.

A few AI-based nutrition tools have already undergone relatively rigorous evaluation. For example, Maher et al. tested an AI-driven virtual health coach in a 12-week randomized controlled trial, demonstrating significant improvements in diet and physical activity adherence [28]. Similarly, Lewis et al. conducted a pilot randomized trial of an AI-enhanced nutrition app in a family setting, reporting improved beverage choices and increased water intake [65]. Although these studies were relatively short in duration and limited in scope, they represent an important step toward high-quality evidence. Building on such examples, better practice would involve the design of multicenter, longer-term randomized controlled trials that assess not only behavioral outcomes (e.g., diet quality, adherence) but also clinical endpoints such as weight, HbA1c, lipid profiles, or cardiovascular outcomes. Embedding these tools within electronic health records and diverse cultural contexts would further strengthen external validity and generalizability.

Table 3 summarizes representative high-quality studies in AI and nutrition, outlining the technique used, the nutritional application, study design, and key findings.

## 6. Research Gaps and Limitations in the Current Literature

Regardless of the rapid expansion of AI applications in nutrition and dietetics, significant research gaps and methodological limitations remain (Table 4). These gaps hinder the translation of promising innovations into reliable, safe, and equitable tools for clinical and public health use.

### 6.1. Lack of Standardized Validation Protocols

One of the most pressing concerns is the absence of standardized protocols for validating AI-based nutrition tools [16]. Current studies vary widely in their evaluation criteria, datasets, and outcome measures, making cross-study comparisons difficult. Without harmonized benchmarks—such as gold-standard reference methods or standardized nutrient databases—it is challenging to assess the true accuracy or clinical utility of AI systems [66]. This heterogeneity in validation approaches also affects regulatory pathways and limits trust among healthcare professionals [43].

### 6.2. Poor Reporting of AI Model Architectures

Another limitation in the existing literature is the inadequate reporting of AI model details. Many studies provide insufficient information about model type, training data, hyperparameters, and update mechanisms. This lack of transparency undermines reproducibility and prevents meaningful peer review [67]. For example, in chatbot-based nutrition interventions, it is often unclear whether responses are rule-based, retrieval-augmented, or generated by LLMs. Without such technical clarity, users cannot fully assess the risks, biases, or strengths of a system [68].

To improve transparency and reproducibility, nutrition-related AI studies could adopt reporting practices already established in data science and biomedical AI. First, structured model cards provide standardized documentation of model architecture, training data, performance metrics, and limitations, making systems more transparent to both researchers and clinicians [69]. Second, dataset datasheets or “nutrition datasheets” could describe the source, demographics, preprocessing, and limitations of training datasets, helping to identify potential biases [70]. Third, widely used frameworks such as CONSORT-AI (for randomized trials of AI interventions), and MINIMAR (for minimum reporting in medical AI) could be adapted to nutrition studies [71,72]. Finally, ensuring that open-source code or at least detailed algorithmic pseudocode is available through repositories such as GitHub or institutional platforms would support independent verification and reuse [73]. Adoption of these structured procedures would bring AI in nutrition research in line with best practices in data science and clinical AI, ultimately increasing trust and reproducibility.

### 6.3. Limited Generalizability Across Populations and Diets

Most AI nutrition tools have been developed and validated within narrow demographic and dietary contexts—primarily among Western, educated, and tech-savvy populations. Consequently, their performance and relevance in more diverse cultural or physiological settings remain uncertain [16,22,27]. Models trained on food logs from North America or Europe may misinterpret traditional meals from Asia, Africa, or Latin America, leading to inaccurate nutrient assessments or inappropriate dietary recommendations [74]. Moreover, language limitations in chatbot-based interventions further restrict applicability outside English-speaking regions [75].

Some encouraging examples of good practice are already emerging. For instance, the International Food Composition Database initiative has incorporated food items from multiple world regions, creating a more representative foundation for AI-based nutrient estimation [76]. Similarly, projects such as the Global Dietary Database and the WHO Global Nutrition Surveillance Network curate dietary intake and health outcomes across diverse countries, providing culturally sensitive data resources [77]. These efforts illustrate how large-scale, inclusive data collection can reduce bias and improve the applicability of AI systems across populations.

### 6.4. Underrepresentation in Low- and Middle-Income Countries (LMICs)

There is a notable scarcity of research and deployment of AI-based nutrition solutions in LMICs. This disparity perpetuates existing health inequities, as populations with the highest burden of malnutrition, food insecurity, and chronic disease are least likely to benefit from emerging digital tools [16,41]. Infrastructure constraints, limited mobile access, lack of culturally tailored content, and absence of localized datasets all contribute to this underrepresentation [78]. More inclusive and context-sensitive research is needed to ensure that AI advances in nutrition support global health equity rather than exacerbate disparities.

### 6.5. Policy and Practice Implications

Beyond research considerations, the rapid adoption of AI in nutrition highlights important implications for healthcare policy. At present, there are no standardized guidelines for evaluating, certifying, or reimbursing AI-based nutrition tools, creating uncertainty for both clinicians and patients. Developing regulatory frameworks that define quality standards, validation procedures, and accountability is essential to ensure safety and reliability [79]. Policies should also promote interoperability with electronic health records and encourage inclusion of culturally diverse dietary data to enhance equity [80]. International initiatives, such as those by the World Health Organization and the European Commission, emphasize the need for governance structures, certification pathways, and ethical oversight for AI in healthcare, and similar efforts should extend to nutrition [21,81]. For professional practice, guidelines from dietetic associations could support the safe integration of AI tools into routine care, clarifying roles and responsibilities of clinicians versus digital systems [13,21,43]. Embedding these considerations into policy would not only foster trust but also accelerate the responsible translation of AI innovations into everyday nutrition and public health practice.

## 7. Conclusions

AI is reshaping nutrition and dietetics by advancing dietary assessment, personalized interventions, public-health planning, and food innovation. Evidence across 2020–2025 shows improvements in efficiency, accuracy, and access—especially when systems integrate sensors, wearables, and multimodal data—yet real-world reliability still varies by context and user adherence. Persistent challenges include algorithmic bias, privacy and security, explainability, and the need for standardized validation and external, multi-site evaluation. In practice, AI should augment—not replace—dietitians; in policy, governance for interoperability, data protection, and transparent reporting remains essential to safe, equitable adoption.

Based on this review’s results, AI is most mature in image-based dietary assessment, personalized coaching/decision support for weight, diabetes, and cardiometabolic risk, and conversational education; these tools show moderate-to-strong accuracy in controlled studies and can improve self-monitoring and engagement. However, generalizability is limited across languages, cuisines, and cultures; model reporting is often opaque; and low- and middle-income country settings are underrepresented, constraining equity and external validity. Priorities are: adopt standardized validation protocols and transparent method reporting; conduct external, multi-site evaluations (including diverse diets and low-resource contexts); and embed AI as clinician-supervised decision support within secure, interoperable workflows.

This review is limited by the heterogeneity of AI methods, datasets, and validation metrics across studies, which precluded direct quantitative synthesis. Our time window and language scope may also underrepresent non-English and very recent work. Evidence gaps persist for underrepresented cuisines and low- and middle-income country settings, pediatrics and older adults, long-term clinical outcomes beyond surrogate markers, implementation and cost-effectiveness studies, and standardized reporting on bias, explainability, privacy, and interoperability. Future research should prioritize externally validated, multi-site trials with diverse diets and populations, common benchmarks and checklists for model reporting, and rigorous evaluations of workflow fit and real-world impact.

Taken together, these findings support cautious but proactive integration of AI as clinician-supervised decision support in nutrition and dietetics, alongside stronger validation and governance.

## Figures and Tables

**Figure 1 healthcare-13-02579-f001:**
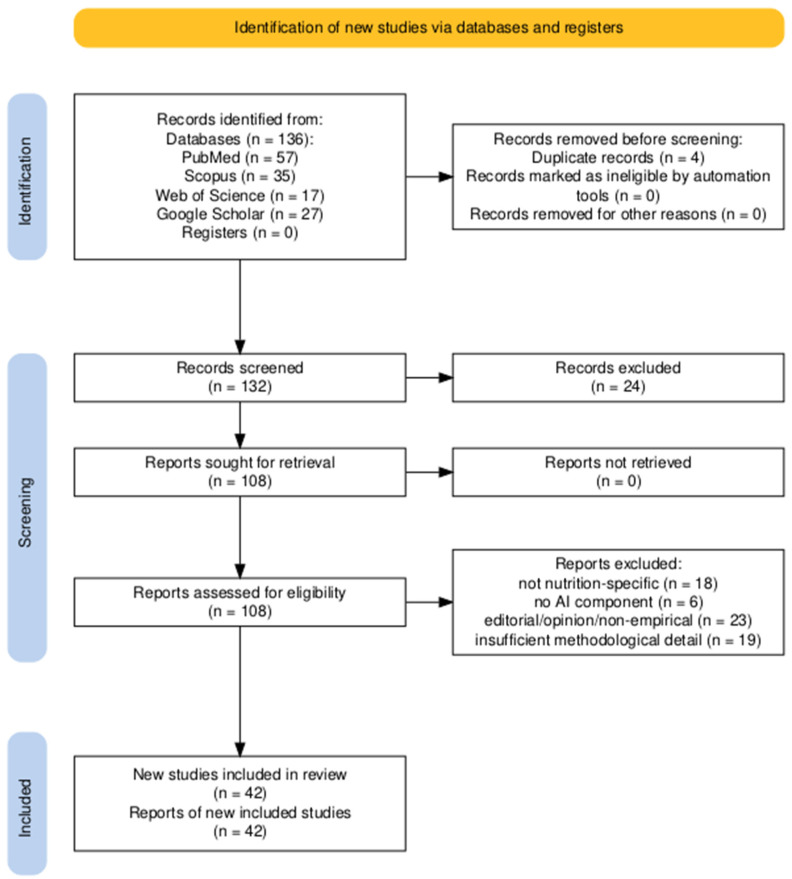
PRISMA-style study selection flow: records identified (by database), duplicates removed, records screened, full-text articles assessed, exclusions with reasons, and studies included.

**Table 1 healthcare-13-02579-t001:** Overview of AI techniques used in nutrition and dietetics (methods-oriented summary).

AI Technique	Application Area	Examples	Strengths	Limitations
Machine Learning (ML)	Dietary assessment, predictive modeling	Prediction of nutrient intake patterns; risk stratification for T2DM/obesity	Learns from large datasets, identifies hidden patterns, adaptable to diverse contexts	Requires large, high-quality datasets; risk of bias from training data
Deep Learning (DL)	Food image recognition, portion estimation	CNN-based food recognition apps (e.g., FoodAI, DietCam)	High accuracy in visual classification; reduces self-report errors	Limited generalizability; depends on food image databases
Natural Language Processing (NLP)	Conversational agents, dietary advice	Chatbots simulating dietitians; LLM-based nutrition Q&A	Enables real-time dialogue, supports education, improves accessibility	Challenges in ensuring accuracy, potential for misinformation
Large Language Models (LLMs)	Virtual coaching, education, health literacy	ChatGPT-based nutrition assistants, personalized diet advice	Generates tailored responses, scalable, user-friendly	Explainability issues, prone to hallucinations, ethical concerns
Reinforcement Learning (RL)	Behavior change support, personalized recommendations	Adaptive diet plans based on user adherence	Learns dynamically from user feedback, supports habit formation	Computationally intensive; limited testing in nutrition contexts
Hybrid Models (ML + Sensors/IoT)	Continuous monitoring, precision nutrition	Wearables integrating HRV, glucose, activity with AI-driven diet feedback	Combines physiological + behavioral data; supports real-time personalized nutrition	Data privacy issues; requires interoperability of devices
Generative AI	Food innovation, flavor design	AI-assisted flavor compound generation, recipe development	Creative potential; accelerates product development in sensory science	Early stage; limited validation of consumer acceptance

Note: Nutrition-specific performance and common failure modes are synthesized narratively in Section 3.1, with representative examples and citations.

**Table 2 healthcare-13-02579-t002:** Ethical and Professional Considerations in AI for Nutrition and Dietetics.

Theme	Key Issues	Implications for Practice	Proposed Strategies
Bias in Training Data	Underrepresentation of certain populations, cultural food diversity not captured	Risk of inaccurate or inequitable recommendations for minority and low-income groups	Curate diverse datasets; conduct fairness audits; continuous model retraining
Transparency & Explainability	Black-box nature of deep learning and LLMs	Reduced trust among clinicians and patients; difficulty in verifying recommendations	Develop interpretable models; provide confidence scores; use explainable AI techniques
Data Privacy & Security	Sensitive dietary, medical, and biometric data at risk	Potential breaches of GDPR/HIPAA compliance; erosion of patient trust	Encryption, federated learning, anonymization, robust consent frameworks
Professional Roles	Concerns that AI might replace dietitians	Threat to professional identity; fear of devaluation of expertise	Promote AI as augmentation rather than replacement; emphasize collaboration
Task-Shifting	Delegation of routine tasks to AI systems	Risk of oversimplifying complex patient cases	Clear delineation of AI vs. human responsibilities; establish clinical oversight
Education & Training Needs	Lack of digital/AI literacy among dietitians and nutritionists	Risk of misuse or over-reliance on AI systems	Integrate AI literacy into curricula and continuing professional education programs
Accountability & Liability	Ambiguity about responsibility when AI advice causes harm	Legal and ethical uncertainty for clinicians and institutions	Define liability frameworks; establish shared accountability between developers & users
Equity in Access	Limited availability in low- and middle-income countries	Risk of widening global health disparities	Promote open-source solutions; support infrastructure development; encourage global policy

**Table 3 healthcare-13-02579-t003:** Key Evidence from High-Quality AI Studies in Nutrition.

Study	AI Technique	Application	Study Design	Key Outcomes
Vasiloglou, 2021 [23]	Deep learning (CNN)	Food image recognition (goFOOD^TM^)	Validation against dietitian assessments	Moderate agreement achieved; main errors in mixed meals and lighting conditions
Wang, 2025 [27]	LLM + image recognition	Personalized meal planning for T2DM	Preclinical validation study	Accurate nutrient analysis and tailored diet recommendations
Maher, 2020 [28]	Virtual health coach (ML + chatbot)	Diet and physical activity counseling	Proof-of-concept RCT (12 weeks)	Significant improvements in fruit/vegetable intake and physical activity
Ponzo, 2024 [35]	AI chatbots (ChatGPT, Bard, Bing)	Dietary advice provision	Comparative evaluation study	ChatGPT most consistent, but all chatbots showed variable accuracy and reproducibility
Chew, 2024 [62]	AI-assisted food tracking app	Eating behavior modification	Mixed-methods evaluation	Improved adherence and user satisfaction with personalized feedback
Lewis, 2023 [65]	AI-enhanced nutrition app	Beverage choice improvement	Pilot RCT	Increased water intake, reduced sugary drink consumption over 3 months

**Table 4 healthcare-13-02579-t004:** Research Gaps and Future Directions in AI for Nutrition and Dietetics.

Research Gap	Current Limitation	Future Direction
Lack of standardized validation protocols	Many studies use inconsistent metrics, making cross-comparison difficult	Develop unified validation frameworks; adopt reporting standards (e.g., CONSORT-AI)
Poor reporting of AI model architectures	Insufficient detail on algorithms, hyperparameters, and training datasets	Encourage transparent reporting; promote open science and model-sharing practices
Generalizability across populations	Most models trained on Western, high-income populations; poor adaptation to diverse diets	Expand datasets to include global populations; integrate cultural dietary variability
Underrepresentation of LMICs	Scarcity of studies and implementation in low- and middle-income countries	Support international collaborations; design resource-appropriate AI solutions
Limited real-world implementation evidence	Many tools tested only in pilot studies or controlled settings	Conduct pragmatic trials and long-term implementation studies in diverse contexts
Data privacy and ethical frameworks	Unclear accountability and uneven compliance with GDPR/HIPAA	Advance federated learning, anonymization, and robust governance frameworks
Integration with clinical workflows	Lack of seamless interoperability with electronic health records (EHRs)	Develop standards for interoperability; test integration in real-world health systems
Equity and access issues	Risk of AI widening health disparities	Design inclusive tools; subsidize access; ensure open-source and low-cost solutions
AI literacy among professionals	Many dietitians lack training in AI technologies	Include AI/ML modules in curricula; create continuing education opportunities
Evaluation of multimodal models	Few studies explore combined use of LLMs, images, and sensor data	Advance multimodal research; test integration with genomics, wearables, and imaging

## Data Availability

No new data were created or analyzed in this study. Data sharing is not applicable to this article.

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
