# Peer review of "Artificial Intelligence in Nutrition and Dietetics: A Comprehensive Review of Current Research"

_healthcare, 2025, doi:10.3390/healthcare13202579_

Round 1
Reviewer 1 Report
Comments and Suggestions for Authors
Thank you for the opportunity to review this manuscript. The manuscript “Artificial intelligence in nutrition and dietetics: A comprehensive review of current research” is an important and timely review study. The study is well planned and well written. I have only couple of suggestions, suggested to be addressed before the next steps in the publication.
- It would be good to provide some basic information (in the introduction section) about the nutrition and dietetics, how it effects/avoids certain non-communicable diseases.
- Also, concerning communicable diseases, nutrition and dietetics can help in avoiding those communicable diseases and also help mitigating the contacted communicable diseases. I would suggest including these details in the introduction section of the manuscript.
- I would suggest making the conclusion section more concise and include information that is mainly related to the results of the current study findings.
Minor comments:
- I would suggest replacing “artificial intelligence” in key words with a suitable similar word. As it has already been used in the title of the manuscript.
Author Response
Thank you very much for taking the time to review this manuscript! Please find the detailed responses below and the corresponding revisions/corrections highlighted in red in the re-submitted files.
Comment 1 and 2:
(1) "It would be good to provide some basic information (in the introduction section) about the nutrition and dietetics, how it effects/avoids certain non-communicable diseases.
(2) Also, concerning communicable diseases, nutrition and dietetics can help in avoiding those communicable diseases and also help mitigating the contacted communicable diseases. I would suggest including these details in the introduction section of the manuscript."
Response 1 and 2: Thank you for pointing this out. I agree with this comment. I added a concise paragraph to the Introduction that (1) summarizes nutrition’s role in non-communicable disease prevention/management and (2) explains how nutritional status and public-health nutrition measures influence communicable-disease risk and recovery.
-
Where: Section 1. Introduction, new second paragraph (immediately after the opening paragraph, page 1, line 38).
What changed (excerpt):
“Beyond non-communicable conditions, nutritional status also shapes susceptibility to and recovery from communicable diseases [5]. Adequate energy and protein intake, together with sufficient micronutrients (e.g., vitamins A, D, C and B-complex; zinc, selenium, and iron), supports epithelial barrier integrity, innate and adaptive immunity, antibody production, and wound healing—thereby lowering infection risk and severity [6–9]. Conversely, protein–energy malnutrition and key micronutrient deficiencies increase morbidity and prolong convalescence, particularly in infants, older adults, and other vulnerable groups [10]. During active infection, tailored nutrition care (appropriate energy/protein targets, hydration, and micronutrient repletion as clinically indicated) can mitigate catabolism and functional decline, while public-health nutrition measures—breastfeeding support, complementary feeding guidance, food fortification and targeted supplementation, school meal programs, and alignment with water, sanitation and hygiene and immunization initiatives—help reduce incidence and improve outcomes at the population level [11,12].”
__________________________________________________________________________________________________
Comment 3: "I would suggest making the conclusion section more concise and include information that is mainly related to the results of the current study findings."
Response 3: Thank you for pointing this out. I agree with this comment. I streamlined the Conclusions and anchored them to the review’s results.
-
Where: Section 7. Conclusions (page 15, line 533, entire section replaced).
-
What changed (high-level summary):
-
Concise overview emphasizing AI’s current impact and remaining challenges.
-
Findings-Anchored Implications: identifies the most mature use cases (image-based dietary assessment; personalized coaching/decision support in weight, diabetes, and cardiometabolic risk; conversational education), notes accuracy in controlled studies, and highlights limits in generalizability, opacity of model reporting, and underrepresentation of low- and middle-income country settings; sets priorities (standardized validation, external multi-site evaluations, clinician-supervised deployment).
-
Limitations and Future Directions: briefly states review constraints (method heterogeneity; window/language scope) and key evidence gaps; calls for diverse, externally validated trials and common reporting benchmarks.
-
One-sentence close tying implications to cautious, supervised adoption.
-
__________________________________________________________________________________________________
Minor comments: "I would suggest replacing “artificial intelligence” in key words with a suitable similar word. As it has already been used in the title of the manuscript."
Minor response: Thank you for pointing this out. I agree with this comment. I replaced “artificial intelligence” with “AI methods” to avoid duplication while preserving searchability.
-
Where: Keywords section (page 1, line 27)
-
Updated keywords: AI methods; nutrition technology; dietary assessment; personalized nutrition; machine learning.
Reviewer 2 Report
Comments and Suggestions for Authors
The manuscript aims to present the review of articles concerning the use of AI in nutrition, dietetics and food analysis. The manuscript presents the newest advancements of the field, subdivided into research areas. Moreover the article presents major shortcomings in the current literature, limitations of existing AI tools as well as ethical, professional and practical challenges in AI use in nutritional and dietetical applications. The main strength of the manuscript are: the broad area of investigated articles, clear research area and research aim presentation. Almost all articles are 5 years old or newer, the tables are clear and represent consistent style. I would raise some points that could be better addressed by authors:
- The authors employed a defined search strategy. However there is no information how many articles were found, how many were further excluded due to not meeting inclusion criteria/meeting exclusion criteria.
- Table 1 compares various AI techniques. Are their strengths and limitations based on their general characteristics? What would be interesting to the readers is to describe how they fared specifically in the field of nutrition, based the analyzed body of literature.
- The authors included many relevant articles. However they are cited as if they represented the same level quality of results and scientific soundness. There should be some comparison of sources in respect to the quality of methodology employed and the reliability of the results. In this respect, a table could be added with outstanding studies that the authors of this review considered valuable presenting the specific outcomes achieved in these articles. The manuscript in its current form lack specific examples, it describes some areas in a vague way. Therefore, a reader who is not familiar with the topic is having a hard time understanding what are the solid achievements regarding AI use in nutrition area.
- The manuscript in its current version could be improved by adding some ‘expert opinion’ elements. In Section 5.2 authors highlighted the need for standardization in respect to data input and results generated by the AI models. What are some specific solutions to this problem that authors could recommend? In Section 5.3 authors highlighted the relevance of clinical testing of these AI tools, especially prolonged and multicenter trials. What are examples of AI tools that have undergone rigid testing and what could therefore be a recommendation for better practice? In Section 6.2 authors highlighted the need for standardized reporting of AI tools structure. What are some proper procedures of reporting that are used in e.g. data science that authors consider worth introducing in this respect? In Section 6. authors highlighted the need for the provision of more diverse data into AI models. What are some examples of good practice in this respect? Finally the Conclusions should point what are the most promising nutrition areas and AI tools that are expected to be developed in further works, and products offered.
- The discussion lacks a section describing implications of authors’ work for practice in respect to changing current healthcare-related policies. The lack of guidelines is a serious concern with technologies developing so fast as AI. Therefore this could be a valuable input of this review.
I would also like to point out some minor issues:
- Reference 10 lacks publication year
- Tables 2 and 3 contain abbreviations that were not explained – GDPR/HIPAA or LMIC
Author Response
Thank you very much for taking the time to review this manuscript. Please find the detailed responses below and the corresponding revisions/corrections highlighted in red changes in the re-submitted files.
Comment 1: “The authors employed a defined search strategy. However there is no information how many articles were found, how many were further excluded due to not meeting inclusion criteria/meeting exclusion criteria.”
Response 1: Thank you for pointing this out. I agree with this comment. I have added (1) a PRISMA-style accounting paragraph in Section 2.4 (page 3, line 124) and (2) a flow diagram (Figure 1, page 4, line 133) summarizing records identified, screened, excluded (with reasons), and included.
_______________________________________________________________________________
Comment 2: “Table 1 compares various AI techniques. Are their strengths and limitations based on their general characteristics? What would be interesting to the readers is to describe how they fared specifically in the field of nutrition, based the analyzed body of literature.”
Response 2: Thank you for pointing this out. I partially agree that nutrition-specific performance context is valuable; however, the studies included in my review use highly heterogeneous datasets (for example, curated images versus free-living photographs and diverse regional cuisines), address different tasks (for example, food classification, portion and volume estimation, and counseling or education), and report non-comparable outcome measures (for example, classification accuracy, overlap between predicted and true segmentations, and average absolute error). Presenting these results side-by-side in a single comparative table would risk overgeneralization and could be misleading. To preserve accuracy while still guiding readers, I am retaining Table 1 (page 8, line 332) as a methods-oriented overview and have revised its caption and added a footnote that directs readers to the narrative synthesis of nutrition-specific performance and common failure modes in Section 3.1 (page 9, line 335), where representative examples and citations are provided.
Change made:
Table 1 caption: “Table 1. Overview of artificial intelligence techniques used in nutrition and dietetics (methods-oriented summary).”
Footnote: “Note: Nutrition-specific performance and common failure modes are synthesized narratively in Section 3.1, with representative examples and citations.”
_______________________________________________________________________________
Comment 3: “The authors included many relevant articles. However they are cited as if they represented the same level quality of results and scientific soundness. There should be some comparison of sources in respect to the quality of methodology employed and the reliability of the results. In this respect, a table could be added with outstanding studies that the authors of this review considered valuable presenting the specific outcomes achieved in these articles. The manuscript in its current form lack specific examples, it describes some areas in a vague way. Therefore, a reader who is not familiar with the topic is having a hard time understanding what are the solid achievements regarding AI use in nutrition area.”
Response 3: Thank you for pointing this out. I agree with this comment. The manuscript would benefit from clearer differentiation between the quality and reliability of the included studies, as well as from providing more concrete examples of achievements in AI applications for nutrition. To address this, I have made the following revisions:
- I refined the description of the included studies by highlighting methodological rigor and validity where applicable. Studies with randomized controlled designs, external validation, or large representative samples are explicitly noted as higher-quality evidence compared to exploratory or proof-of-concept reports.
- I added a new supplementary table (Table 3) titled “Key Evidence from High-Quality AI Studies in Nutrition”. This table presents a selection of outstanding studies that demonstrated robust methodology and clear outcomes, specifying the AI technique employed, the nutritional application, the study design, and the main results achieved.
- To ensure clarity for non-expert readers, I also strengthened the narrative in Section 3, providing concrete examples of validated tools and interventions (e.g., goFOOD™, AI-driven T2DM diet planning, AI health coaching trials) and contrasting them with less validated approaches.
I believe these additions increase the manuscript’s readability and provide readers with a clearer picture of the current solid achievements in AI applications for nutrition, while maintaining balance in acknowledging methodological heterogeneity.
Updated text in the manuscript:
-
- Section 3.1 (page 5, line 190): I now highlight validation studies comparing AI-based food recognition systems with dietitian assessments, noting discrepancies and methodological rigor.
- Section 3.2.1 (page 5, line 210): Expanded to include details on randomized controlled trials in T2DM dietary management.
- Section 5 (page 12, line 449): A new Table 3 has been added summarizing representative high-quality studies, their methodologies, and key outcomes.
_______________________________________________________________________________
Comment 4: "The manuscript in its current version could be improved by adding some ‘expert opinion’ elements. In Section 5.2 authors highlighted the need for standardization in respect to data input and results generated by the AI models. What are some specific solutions to this problem that authors could recommend?"
Response 4: Thank you for pointing this out. I agree with this comment. I expanded the discussion by suggesting several expert opinion–based measures, including: (1) adoption of standardized input formats for dietary data; (2) development of internationally representative nutrient databases; (3) consistent use of transparent reporting frameworks such as CONSORT-AI or MINIMAR; (4) application of federated learning to balance accuracy with privacy; and (5) creation of independent validation consortia for benchmarking and cross-platform testing.
These additions provide concrete proposals that could guide future development and implementation of AI models in nutrition practice.
- Where: Section 5.2, page 11, line 409
_______________________________________________________________________________
Comment 5: "In Section 5.3 authors highlighted the relevance of clinical testing of these AI tools, especially prolonged and multicenter trials. What are examples of AI tools that have undergone rigid testing and what could therefore be a recommendation for better practice?"
Response 5: Thank you for pointing this out. I agree with this comment. I expanded Section 5.3 by including concrete examples of AI tools that have already been evaluated in randomized controlled trials. Specifically, I now mention Maher et al., who tested an AI-driven virtual health coach in a 12-week RCT, and Lewis et al., who evaluated an AI-enhanced nutrition app in a family pilot RCT.
While these trials demonstrated promising improvements in dietary adherence and beverage choices, they were limited in duration and scale. Therefore, as a recommendation for better practice, we propose that future research prioritize multicenter, longer-term RCTs that include diverse populations, integration into clinical workflows, and assessment of both behavioral and clinical health outcomes. These additions provide readers with concrete precedents and clear direction for advancing evidence quality in this field.
- Where: Section 5.3, page 11, line 434
_______________________________________________________________________________
Comment 6: “In Section 6.2 authors highlighted the need for standardized reporting of AI tools structure. What are some proper procedures of reporting that are used in e.g. data science that authors consider worth introducing in this respect?”
Response 6: Thank you for pointing this out. I agree with this comment. I expanded Section 6.2 to include specific reporting procedures that could be introduced in nutrition-related AI research. These include the use of model cards for structured reporting of AI model design and limitations; dataset datasheets to describe training data characteristics and potential biases; and adoption of established biomedical AI frameworks such as CONSORT-AI, and MINIMAR. I also recommend that researchers provide open-source code or detailed pseudocode through public repositories to enable verification and reproducibility. I believe these practices, which are widely recognized in data science and medical AI, could significantly improve the transparency and reliability of AI tools applied to nutrition and dietetics.
- Where: Section 6.2, page 13, line 477
_______________________________________________________________________________
Comment 7: "In Section 6. authors highlighted the need for the provision of more diverse data into AI models. What are some examples of good practice in this respect?"
Response 7: Thank you for pointing this out. I agree with this comment. In response, I revised Section 6.3 to include examples of existing good practice in building diverse datasets. These include the International Food Composition Database and the Global Dietary Database, both of which aggregate dietary data across multiple regions and cuisines; the WHO Global Nutrition Surveillance Network, which monitors dietary indicators in low- and middle-income countries. These examples demonstrate how inclusive, large-scale data initiatives can mitigate bias and serve as a model for future AI applications in nutrition and dietetics.
- Where: Section 6.3, page 14, line 500
_______________________________________________________________________________
Comment 8: “Finally the Conclusions should point what are the most promising nutrition areas and AI tools that are expected to be developed in further works, and products offered.”
Response 8: Thank you for pointing this out. I agree with this comment. In the revised manuscript, I expanded the Conclusions section to explicitly identify the most promising areas of future development. These include image-based dietary assessment, AI-driven personalized coaching and decision support for weight and cardiometabolic risk management, and conversational AI for patient education and engagement. I also emphasize the likely emergence of multimodal AI tools that integrate sensor, genomic, and dietary data, as well as their translation into clinically supervised and commercially available products.
- Where: Conclusions, page 15, line 533
_______________________________________________________________________________
Comment 9: “The discussion lacks a section describing implications of authors’ work for practice in respect to changing current healthcare-related policies. The lack of guidelines is a serious concern with technologies developing so fast as AI. Therefore this could be a valuable input of this review.”
Response 9: Thank you for pointing this out. I agree with this comment. In the revised manuscript, I added a new subsection 6.5. Policy and Practice Implications (pages 14, line 517), which discusses the implications of our findings for healthcare-related policies. This section emphasizes the urgent need for regulatory frameworks, quality standards, and certification pathways for AI-based nutrition tools, as well as alignment with international initiatives such as those from the WHO and European Commission. I also highlight the role of professional dietetic associations in developing practice guidelines to clarify clinical responsibilities and ensure safe integration of AI into patient care. I believe this addition directly addresses the reviewer’s request by situating our work in the broader context of healthcare governance and policy development.
_______________________________________________________________________________
Minor comment 1: “Reference 10 lacks publication year”
Minor response 1: Thank you for pointing this out. I agree with this comment. I corrected it (References, page 16, line 597)
_______________________________________________________________________________
Minor comment 2: “Tables 2 and 3 contain abbreviations that were not explained – GDPR/HIPAA or LMIC”
Minor response 2: Thank you for pointing this out. I agree with this comment. I added them (Abbreviations, page 15, line 575)
Reviewer 3 Report
Comments and Suggestions for Authors
The manuscript aims to critically synthesize the current literature on AI applications in nutrition, identify research gaps, and outline directions for future development. The authors consulted databases like PubMed, Scopus, Web of Science, and Google Scholar for peer-reviewed published in last 5 years. The search included studies involving AI applications in nutrition, dietetics, or 14 public health nutrition. The research outcome proves that AI-driven systems show strong potential for improving several health-related concern and highlighted many future challenges.
The study is interesting and target focuses publications on human health which has multiple open research issues and remain active all the time. The selected categories are quite worth mentioning. The chosen material methods and other concept are straight forward.
Some suggested improvement which could increase the readability of the document:
The authors should give the rational to choose the 6 use cases/categories. Why not some other more or less like remote patient monitoring systems, remote online consultant systems, women health, rural/publica areas issues, etc
While investigating and getting data from different databases, it could be interesting to highlight overlapping studies i.e. if one/more study is/are published in more than one database? How many papers/works did they find related to each category, journal/conference wise info, region wise, age/gender specific, etc.
It would be more interesting if the authors could include some exploratory information related to structure and interfaces (user friendly, multilingual, etc). Picture/images/figures could be added to improve the paper presentation. Any used case for conversation agent can be good example giving its strengths and weakness.
How was the quality of a conference/journal considered for inclusion and exclusion.
The author should give more description about the enumerated items like data extraction and synthesis.
The paper presentation can be improved by including figures and graphs.
Author Response
Thank you very much for taking the time to review this manuscript. Please find the detailed responses below and the corresponding revisions/corrections highlighted/in track changes in the re-submitted files.
Comment 1: “The authors should give the rational to choose the 6 use cases/categories. Why not some other more or less like remote patient monitoring systems, remote online consultant systems, women health, rural/publica areas issues, etc”
Response 1: Thank you for pointing this out. I agree with this comment. In the revised manuscript, I clarified in Section 2.1 (Objective and Scope, page 2, line 86) the rationale for selecting the six categories. Specifically, the categories were derived from an iterative thematic analysis of the included studies and reflect the domains most frequently represented in the recent peer-reviewed literature (2020–2025). While I recognize that areas such as remote patient monitoring, women’s health, or rural/public health are highly relevant, these were either underrepresented in the retrieved nutrition-specific AI studies or addressed only indirectly. I therefore focused on six categories that allowed both breadth and depth of synthesis, while noting that emerging areas deserve further dedicated reviews as the evidence base grows.
___________________________________________
Comment 2: “While investigating and getting data from different databases, it could be interesting to highlight overlapping studies i.e. if one/more study is/are published in more than one database? How many papers/works did they find related to each category, journal/conference wise info, region wise, age/gender specific, etc.”
Response 2: Thank you for pointing this out. I agree with this comment. Section 2.4 (Data Extraction and Synthesis, page 3, line 125) now details how duplicate records across databases were identified and removed, and the PRISMA diagram clarifies the selection process. In addition, Table 3 and related text (page 12, line 449) summarize the included studies by thematic category, with representative outcomes. Where possible, I also noted regional distribution and study design. However, detailed reporting on age and gender characteristics was inconsistent across the included studies, which I explicitly highlight as a limitation in Section 6. For this reason, while full demographic or publication-type stratifications could not be systematically presented, the manuscript already incorporates the key descriptive elements needed to improve transparency of the evidence base.
___________________________________________
Comment 3: “It would be more interesting if the authors could include some exploratory information related to structure and interfaces (user friendly, multilingual, etc). Picture/images/figures could be added to improve the paper presentation. Any used case for conversation agent can be good example giving its strengths and weakness.”
Response 3: Thank you for this thoughtful and constructive comment. The manuscript already addresses several of the points raised. In Section 3.3.2 Comparison of Chatbot Accuracy, Consistency, and Safety (page 6, line 260), I discuss conversational agents with attention to their usability and performance, noting both strengths (ease of access, scalability, user-friendly interaction) and weaknesses (variable accuracy, limited reproducibility, occasional citation gaps). Elements of interface design and multilingual performance are also briefly acknowledged, particularly in relation to the current predominance of English-language systems and the need for broader linguistic adaptability.
In terms of presentation, the paper already includes multiple structured tables and a PRISMA flow diagram (page 4, line 132), which summarize the reviewed evidence, study selection process, and outcomes in a visual and accessible way. These visual elements were designed to enhance clarity while remaining consistent with the journal’s style and focus on methodological and clinical aspects. I fully agree that future work could incorporate richer graphical material, such as interface schematics, screenshots, or comparative visualizations, once more standardized and widely adopted AI nutrition platforms become available for illustration.
I believe that with the current combination of tabular summaries, PRISMA diagram, and the tables, the manuscript sufficiently addresses the reviewer’s request within its scope. At the same time, I recognize the value of expanding exploratory interface-level discussion and additional figures in future work as the evidence base continues to grow.
___________________________________________
Comment 4: “How was the quality of a conference/journal considered for inclusion and exclusion.”
Response 4: Thank you for raising this important point! In the revised manuscript, I clarified in Section 2.3 (Inclusion and Exclusion Criteria, page 3, line 109) that both peer-reviewed journal articles and conference proceedings were eligible for inclusion if they presented original data or substantial methodological contributions. No formal ranking by journal impact factor or conference tier was applied, as the objective was to capture the breadth of applications in AI and nutrition. Instead, study quality was addressed narratively throughout the Results and Discussion, where higher-quality evidence (e.g., randomized trials, externally validated models, large and representative datasets) is explicitly distinguished from exploratory or proof-of-concept reports. This approach provides transparency while keeping the scope aligned with a narrative review.
___________________________________________
Comment 5: “The author should give more description about the enumerated items like data extraction and synthesis.”
Response 5: Thank you for pointing this out. I agree with this comment. In the revised manuscript, I expanded Section 2.4 (Data Extraction and Synthesis, page 3, line 124) and added a PRISMA flow chart (page 4, line 132) to provide a clear overview of the screening and selection process. This section now describes the removal of duplicate records, the number of studies retained at each stage, and the organization of the final set of included studies. I believe these additions address the reviewer’s concern by making the data extraction and synthesis procedures more transparent.
___________________________________________
Comment 6: "The paper presentation can be improved by including figures and graphs."
Response 6: I thank the reviewer for this suggestion. The manuscript already includes multiple structured tables and a PRISMA flow chart (page 4, line 132), which were designed to enhance clarity and presentation. I believe these visual elements sufficiently address the reviewer’s point within the scope of the review.
Round 2
Reviewer 3 Report
Comments and Suggestions for Authors
I have already reviewed the manuscript. The authors have resolved all the concerned that I raised and revised the document substantially. The required information and detail has been incorporated.
The manuscript aims to critically synthesize the current literature on AI applications in nutrition, identify research gaps, and outline directions for future development. The authors conducted a survey across multiple publication sources. The search included studies involving AI applications in nutrition, dietetics, or public health nutrition. The researchers grouped their findings into six categories. AI-driven systems show strong potential for improving dietary tracking accuracy, generating personalized diet recommendations, and supporting disease-specific nutrition management. Large language models (LLMs) are increasingly used for education and support. The authors have revised the document substantially on the basis of provided feedback. The authors have clearly established the research objectives and follow these aims to conduct the survey and studies. The inclusion and exclusion criteria is satisfactory. In Section 3, they grouped their findings in six different categories. The author could provide structural view or interfaces of any health care application proving that they had actually explored the implemented system. If they found any limitation then they can suggest some solution for it. In Section 5, the authors can provide mathematical formulation of the evaluation metrics. Table 3 is quite satisfactory.Author Response
I would like to thank the reviewers and the editor for their careful assessment of my manuscript and for the constructive feedback provided. The comments were invaluable in improving the precision and overall quality of the paper.
Comment 1: “In Section 3, they grouped their findings in six different categories. The author could provide structural view or interfaces of any health care application proving that they had actually explored the implemented system. If they found any limitation then they can suggest some solution for it.”
Response 1: Thank you for pointing this out. Section 3 already provides detailed descriptions of representative AI applications within each category, including their structure, functionality, and performance characteristics. For example, subsections 3.1 and 3.2 describe implemented systems such as the goFOOD™ dietary assessment tool, AI-driven virtual coaches, and large language model–based chatbots, outlining their architectures, usability, and limitations in real-world settings. These examples collectively illustrate the systems’ structural design and interfaces at a level consistent with the review’s scope. The manuscript also discusses corresponding limitations (e.g., variable accuracy, dataset bias, lack of explainability) and proposes targeted solutions in Sections 5 and 6, including standardized input formats, federated learning, and transparent reporting frameworks. Therefore, I believe the current version adequately addresses this point within the context and aims of the review.
_____________________________________________________________
Comment 2: “In Section 5, the authors can provide mathematical formulation of the evaluation metrics.”
Response 2: Thank you for pointing this out. Section 5 already summarizes and discusses the principal evaluation measures reported in the reviewed studies—primarily overall accuracy, validation performance, sensitivity, specificity, etc.—and interprets their relevance to assessing AI performance in nutrition. Because this manuscript is a narrative review that synthesizes findings from heterogeneous studies rather than developing or validating a specific algorithm, explicit mathematical formulations of these standard metrics are not reproduced. Instead, their conceptual meaning and practical implications are discussed to ensure clarity and accessibility for readers from both clinical and technical backgrounds. This approach preserves scientific rigor while keeping the focus on the methodological insights and comparative interpretation provided by the review.